# Social expectations and government incentives in Malaysia's COVID-19 vaccine uptake

**N. Izzatina Abdul Aziz**[1,2☉]*, **Sam Flanders**[2,3,4☉], **Melati Nungsari**[2,3,4☉]

1 Institute of Malaysian and International Studies (IKMAS), National University of Malaysia, Bangi, Selangor, Malaysia, 2 ASEAN Research Center, Asia School of Business, Kuala Lumpur, Malaysia, 3 Asia School of Business, Kuala Lumpur, Malaysia, 4 MIT Sloan School of Management, Cambridge, MA, United States of America

☉ These authors contributed equally to this work.
* izzatina.aziz@ukm.edu.my

**Data Availability Statement:** Data are available on the Open Science Framework (https://osf.io/jtc5q/).

**Funding:** The author(s) received no specific funding for this work.

## Abstract

High vaccination rates are integral to reducing infection and severity rates of COVID-19 infections within a community. We examine the role of social expectations in COVID-19 vaccination take-ups and its interaction with potential government actions in Malaysia. We find that individuals' expectations of others in their social groups towards vaccination predicts those individuals' vaccination registrations. Using a vignette experiment, we examine the extent of normative expectations in normalizing pro-vaccination behavior beyond an individual's reference group. We find that unless moderated by a high level of public trust, individuals prefer punitive policies as a way to increase vaccination rates in their communities.

## Introduction

Vaccinations against the SARS-CoV-2 virus have been identified as an effective medical intervention that reduces the risk of infection, transmission, and severity from COVID-19. Incentivizing booster uptakes for future waves and future variants will be an ongoing challenge, with fourth doses now being offered in many countries worldwide [1–3]. Aside from the private benefits of vaccination, individuals who choose to get vaccinated also generate positive externalities by reducing community transmission and they also improve the ability of the healthcare system to respond to non-COVID-19 related illnesses [4]. Thus, increasing COVID-19 vaccination is an ongoing policy goal for many governments.

Most COVID-19 vaccination policies around the world have relied on voluntary take-ups rather than mandating vaccinations for the whole population. However, policymakers faced the lingering problem of general vaccine hesitancy and resistance being carried over to COVID-19 vaccine uptake. Individuals could object to vaccination altogether (being resistant to the vaccine), or hesitate to take up the vaccine as they are unsure about the benefit of vaccination [5, 6]. This makes vaccination a *social dilemma*, in which much of the benefit does not accrue to the individual making the choice. Vaccination decisions could stem from

**Competing interests:** The authors have declared that no competing interests exist.

individuals' beliefs that vaccinations will benefit them directly, or they could be affected by decisions made by others around them.

Policymakers can encourage vaccination by providing information [7], granting benefits such as small cash incentives [8–10] or limiting access to physical spaces based on vaccination status, or even mandating vaccination [11]. While nudge interventions are hypothesized to be cheap and quick to implement, their impact has been highly mixed [8, 9, 12–15], as have the impact of lotteries in increasing vaccination uptakes [9, 13].

Individuals' decisions to get vaccinated are also a function of decisions of "relevant others", i.e. external parties whose opinions or social relations matter to said individuals. Possible channels for this include i) social norms ii) herding, and, iii) externalities and free-riding behaviour. With herding, individuals assume others may know more than they do, and mimic them for that reason. Free-riding occurs when individuals assume that others will solve the social dilemma and conclude they don't need to contribute (i.e. get vaccinated) to reap the benefits.

In this paper, we focus on social norms. Our study investigates the influence of social norms by examining the *empirical* and *normative* expectations along with its interactions with government interventions on vaccination decision-making. Recent works during the COVID-19 pandemic have demonstrated the role of social norms and their importance in ensuring the efficacy of COVID-19 interventions like vaccination uptake [16, 17], ensuring social distancing compliance [18] and also how to inform COVID-19 vaccination policy design [19]. In the social norms framework [20, 21], individuals voluntarily adopt behaviors based on what they believe others think they ought to do, which itself may be inferred from others' behavior. We also included measure of trust in government agencies and its potential policies in normalizing COVID-19 vaccination take-up in the society.

As there is significant cross country variation in attitudes towards vaccination [22, 23], so answers to these questions cannot reliably be generalized, but must be studied in a local context–in our case, Malaysia. Our study also examine actual vaccination behavior, i.e. vaccination registration, while majority of works from Malaysia focus on stated knowledge, attitude and practice, as well as reported acceptability. Currently only [24] examines actual vaccination behavior via vaccination registration through a natural experiment of small cash payments for voters in a constituency, and the opt-in policy for populations of certain states to register for the 'less desirable' AstraZeneca vaccine [25].

## Materials and methods

### Settings

In Malaysia, an upper middle income Southeast Asian nation of approximately 32 million residents, the National COVID-19 Immunisation Programme (PICK) is managed by the COVID-19 Immunisation Task Force (CITF), which is responsible to procure, arrange for logistics, and manage the registration for all Malaysian residents to receive the COVID-19 vaccine. The task force was established on the 20th of January, 2021 under the purview of the Malaysian Ministry of Health (MoH) and Ministry of Science, Technology and Innovation. The roll-out of the vaccines was done in stages, beginning 24 February 2021. Phase 1 targeted 'frontliners' and involved 500,000 individuals while Phase 2 began in April 2021, prioritising senior citizens (aged 60 years and above), and high-risk individuals with severe disease(s) and disabilities. This involved around 9.4 million individuals or 28.8% of the population.

The remaining population aged 18 years and above were to receives their vaccination under Phase 3. Those under Phase 3 were prioritized by age and location (with high disease burdens) to help contain infection and lessen the burden on healthcare facilities. Unlike many other countries with slow starts to their vaccination programs, the national demand for vaccines

outstripped its coverage and rates from 1 April 2021 to 15 September 2021 [26]. This high demand is consistent with the finding from [23] that Malaysian respondents have higher COVID-19 vaccination acceptance rate compared to respondents in Middle East and Europe.

There were several channels to register for vaccinations through PICK—one could, for example, walk in to public clinics and hospitals to sign up, or go to several mobile registration sites. However, the main channel was through MySejahtera, the dominant app for national contact tracing in the country, which was re-purposed to also accommodate vaccine registrations. The app is used widely since contact tracing and location reporting were made mandatory starting 1st of August, 2020. The introduction of vaccine registration into MySejahtera has made the registration process seamless while removing potential administrative burdens on citizens.

The government also guaranteed the public that vaccination through PICK will be free irrespective of citizenship status. Everyone was guaranteed to receive a full regime of COVID-19 vaccine shots (either 1 or 2 doses, according to the type of vaccine), following the time frame set by the medical professionals in the Ministry of Health, Malaysia. However, due to perceived slowness of the PICK program and the necessity of some economic sectors to continue operating, several parallel vaccination programs were introduced by several government bodies. Organisations and individuals that decided to go through this alternative channel had to pay for their COVID-19 vaccines. As of October 2021, Malaysia had provided emergency/conditional approvals to 7 types of COVID-19 vaccines but only 5 of them were distributed to the public. The most widely distributed vaccines in Malaysia are Comirnaty (Pfizer/BioNTech), Vaxzevria (Oxford/AstraZeneca) and CoronaVac(Sinovac). These vaccines are either in a form of mRNA, adenoviruses or inactivated-virus and trials were conducted before the vaccines were approved for worldwide distribution to ensure their efficacies in reducing incidences and severity of illnesses of COVID-19 infections [27, 28].

It is important to note that the overall vaccination acceptance under the National Immunization Program (NIP) in Malaysia through free immunization of children for non-COVID-19 related illnesses was widely accepted. In 2020, the coverage of childhood vaccinations reached more than 95% of its target immunization. For example, the BCG immunization coverage nationally is at 98.8% and the HPV immunization for girls at 13 years old is at 95.73% [29].

In recent years, there have been increasing signs of anti-vaccination and vaccine-hesitancy behaviours in Malaysia, reflected by the reported increase of vaccine-preventable diseases [30–32]. This includes reported incidences and deaths stemming from vaccine-preventable childhood diseases like measles, diphtheria and pertussis among non-vaccinated individuals and the reluctance of young adults to receive the HPV vaccine. [30, 33] discussed several potential reasons behind vaccine hesitancy in Asian and Malaysian contexts and showed that apart from past experience(s) and adverse events following immunization (AEFI) and the role of social media in spreading anti-vaccination rumours, local contexts matter. For example, the prevalence of traditional complementary and alternative medicines (TCAM) usage in Malaysia can misconstrue vaccination benefits, and the lack of clear ruling from Islamic authorities shapes the perception of the permissibly or 'halal' status of vaccination consumption [32–34]. These pre-COVID-19 anti-vaccination and hesitancy behaviors indicated there could be potential resistance faced by policymakers in immunizing the population during the COVID-19 pandemic.

A novel coronavirus (COVID-19) was discovered to be the source of clusters of respiratory illness cases in Wuhan, China, in late 2019 [35]. Before the presence of vaccines, the focus of policymakers was to reduce the spread of COVID-19 using non-medical interventions. Since spread of COVID-19 virus among humans could happen via contact and airborne transmission and the onset of illness only appears a few days after contact [35], the use of physical

distancing, face masks and eye protection have positive effects in reducing the risk of contracting the COVID-19 virus [36]. Most governments rely on citizens' willingness to change their behavior and comply with restrictions from the government. Channels that facilitated the population's compliance with behaviors that prevented COVID-19 spread included the belief that recommended behaviors were scientific, and faith in government's ability to handle the pandemic [18, 37]. In the Malaysian scenario, [37–41] documented the general population's knowledge and awareness of COVID-19 measures and a willingness to change behaviors at the early stage of the pandemic.

Several studies have documented the acceptability of COVID-19 vaccination in Malaysia. [42] reported the majority of their Malaysian respondents (96%) would accept vaccination as an effective control measure, revealing also that the convenience of obtaining the vaccine and doctors' recommendations had the ability to influence their decisions. They also examined socio-demographic factors behind vaccination acceptance and found that age, race, income level, education level have associations with willingness to accept vaccination. Since Malaysia is a Muslim-majority country, [43] focused on the acceptability of COVID-19 vaccination and also measured respondents' fatalistic health beliefs. The majority of respondents expressed a willingness to get vaccinated although there were attitudinal barriers preventing its unconditional acceptance, such as acceptance of the vaccine's halal status, the presence of non-vaccine alternatives, and fatalistic health beliefs which accept illness occurring in one's family as a given.

## Study design

An anonymous, self-administered and cross-sectional online survey was conducted in Malaysia for this project. The survey was administered using Qualtrics. The survey had 6 major blocks. In Blocks 1 to 4, all respondents answered questions about themselves, their expectations of vaccination and herd immunity, as well as several dimensions of trust. The type of questions that respondents received in Block 5 depended on their self-reported vaccination registration status and their willingness to share their contact details with the researchers. Respondents who were not registered with PICK but were willing to provide contact details and identification numbers were sorted into a lottery experiment. This aimed to examine the role of cash incentives to encourage individuals to take up vaccination. Respondents who had registered were classified as either an immediate adopter or a late registrant for vaccination, and received different questions in Block 5. Finally, unregistered individuals who were unwilling to provide contact details were asked another set of questions in Block 5. All respondents other than lottery experiment participants answered questions in Block 6, the vignette experiment. The survey questions can be found in S1 Appendix. Translated survey questions along with its relevant data and codes are currently hosted at Open Science Framework database and can be accessed at https://osf.io/jtc5q/.

This project was conducted under the overview of the Asia School of Business (ASB)'s Institutional Review Board, (IRB No: ASB-IRB-2021–3) and was approved prior to the dissemination of the survey.

## Respondents

The survey was open to all Malaysian residents above the age of 18 years (note: not only citizens, but every resident within national borders). Respondents had the option of answering the survey in three languages commonly used in Malaysia: English, Malay, and Mandarin. Respondents were recruited using snowball sampling by advertising through ASB's social media platforms and circulating of the Qualtrics link through business and personal messaging

networks such as email and WhatsApp. We also pro-actively engaged with key opinion leaders and non-governmental organizations to increase the number of respondents. Data collection took place from June 21 to July 5, 2021. The data collection period coincided with the mass roll out of COVID-19 vaccinations Malaysia. The survey was completed by 1,307 respondents —1,017 in English, 203 in Malay, and 87 in Mandarin. Filtering and consolidating survey data leads to 1,297 valid respondents.

## Survey design

Fig 1 describes the flow of the survey.

After respondents answered questions about themselves in Block 1, they reported their empirical expectations on the vaccination take-up in several types of reference groups. Respondents reported the percentage of individuals in their reference groups who they believed had registered to be vaccinated. We distinguished the reference groups based on several categories, i) the members of respondents' household, ii) their extended family, iii) their friends, iv) their co-workers, v) their neighbours and iv) members of their religious group. Measuring respondents' expectations at different types of social groups levels enabled us to narrow relevant groups that served as reference groups that have and can influence pro-vaccination behaviour among respondents. This is consistent with the social norms framework of [20] and we hypothesised that respondents that have high pro-vaccination expectations of their reference group(s) are more likely to be registered; eventually get vaccinated and they are also more likely to influence others, or be influenced by others to be vaccinated.

Next, respondents reported their levels of trust of different social groups and relevant government branches and actors that are important in projecting confidence towards COVID-19 vaccines as a medical intervention for Malaysian society. In Block 4, we show respondents an infographics that illustrates the concept of herd immunity. Below it, we elicited respondents' expectations of vaccination registration, herd immunity and peer support for any future government incentive for vaccination or punishment for vaccination refusal. The infographs that were showed torespondents are similar to the infographs produced by the British Society for Immunology [44].

The majority of respondents completed questions in Block 6. They were asked three questions about an unregistered protagonist living in a country that has a clearly specified policy goal of achieving herd immunity. Individuals were asked to imagine the protagonist's

| Block 1 | Block 2 | Block 3 | Block 4 | Block 5 | Block 6 |
|---|---|---|---|---|---|
| Demographics | Vaccination Expectations | Trust | Herd immunity expectations | Actual vaccination behavior | Vignette |
| | | | | A. Early adopter registered respondents<br><br>Q29 – Q35 + Q36 | |
| | | | | B. Eventually registered respondents<br><br>Q29 -Q35 + Q37-Q38 | 3 vignette questions |
| Q2 – Q9 | Q10-Q12 | Q13-Q14 | Q16 – Q21 | C. Unregistered respondents<br><br>Q29 – Q30 | |
| | | | | D. Unregistered & sorted in lottery experiment | |

**Fig 1. Survey flow.** Respondents answered questions in Block 1 to Block 5 before reporting their vaccination registration status. Individuals who reported their willingness to share personal information and were yet to be registered into the national vaccination program were sorted in an experiment. Personal information was excluded and destroyed in data analysis process. The remaining respondents—i.e. those who had registered for vaccination, those who had been vaccinated, and unregistered individuals who were not willing to share personal information- proceeded to the parts of the survey that explored their reasons for taking up or refusing vaccination. These respondents then randomly sorted into 1 of 4 different vignettes. In each vignette, respondents were provided their expectations on a third party's vaccination behaviour and government policies to increase take-up levels of vaccination were varied.

likelihood to register to be vaccinated, to actually get vaccinated, and whether vaccination was the appropriate prescribed action. These predictions were solicited with a five point Likert scale, but were converted to a binary scale in our analysis, with "neutral" to "extremely unlikely" as a negative outcome and "likely" to "extremely likely" as a positive outcome. The description of the situation is as follows: "Achieving herd immunity is a goal of many governments in the world. Imagine that somebody like you lives in Malaysia and hasn't registered for the vaccine. In this scenario, most residents in Malaysia *have refused to register/have registered* for the vaccine. *The government is providing an incentive to get registered and vaccinated by conducting a nationwide lucky draw to win RM1 million / The government is punishing residents by requiring that non-vaccinated residents must pay for PCR Covid tests every 14 days in order to work in person*. The manipulation is expressed in italics. This methodology is adapted from [18] that used a vignette experiment to examine individual's compliance with social distancing rules. [18] uses two manipulations of social expectations: whether other people are complying or not (*empirical expectations*) and whether other people support compliance or not (*normative expectations*). We add the incentive valence manipulation described above, and, to avoid cutting our sample into eight, compress the expectations manipulation into a single empirical expectations manipulation under the assumption that, in the absence of additional information, respondents will infer that widespread compliance implies higher support for registration and low compliance implies low support for registration.

The respondents who entered Block 6 were sorted in a vignette with low population registration (LP) or high population registration (HP). Varying the level of population registration allowed us to simulate a condition in which vaccination expectations are either high or low within a community. Within vaccination prevalence categories, some respondents were sorted into a policy context that had the government incentivizing them through lottery (LOT), and disincentivizing the unvaccinated by requiring them to take frequent PCR tests (PUN). Table 1 summarizes the vignette's treatments.

The minority of respondents who did not enter Block 6 were involved in an incentivized experiment. In this small experiment, the treated respondents were provided information on a lucky draw worth RM500 (USD120) if they decided to register to be vaccinated before a deadline. Those sorted in the control group only received a message thanking them for participation. Our research team used the provided details to inspect whether registration did take place in the PICK system. Personal details for this category of respondents have been destroyed and excluded from the data analysis process.

## Statistical analysis

Statistical analysis was performed using STATA [Version 15]. Descriptive analyses were used to describe respondents' characteristics and the summary statistics for the main variables of interest; namely vaccination expectations and trust. Appropriate inferential analyses implemented to examine normalization of vaccination in Malaysian society and its interactions with trust and potential policy actions. Respondents' answers in Block 6 serve as variables of interest to examine normalization of COVID-19 vaccine takeup.

**Table 1. Vignette treatments.**

| | Vaccination Incentives | |
|---|---|---|
| Prevalence of registration | Low registration (LP), Punishment for refusal (PUN) | Low registration (LP), Lottery entry as reward (LOT) |
| | High registration (HP), Punishment for refusal (PUN) | High registration (HP), Lottery entry as reward (LOT) |

The aim of the inferential analyses is to examine respondents' expectations towards the protagonist's behaviour based on the treatment assigned. Under different levels of societal expectation of vaccination, (HP vs. LP), we hypothesised that respondents' likelihood to believe that the protagonist will register and get vaccinated will vary based on treatment. Respondents assigned in HP are more likely to predict protagonist vaccination take-up compared to respondents assigned to the LP, as there is high social pressure/expectation that others will take-up vaccination. This will be the Hypothesis 1 (**H1**). As the vignette also incorporates potential policies faced by citizens with respect to vaccination decisions, the second hypothesis that we will test (**H2**) is the relative efficacy of reward versus punishment under both levels of vaccination prevalence. Subsequently, we will examine how respondents' empirical expectations and trust level translated into their belief that the protagonist is likely to register, get vaccinated and whether they should be vaccinated.

$p$-value of less than .05 was considered to be statistically significant. For analysis that compares expectation between unregistered and registered respondents, we will use $q$-value as a way to control the false discovery rate as it is unlikely that we will obtain a balanced set of unregistered and registered respondents. Similar to $p$-value, the $q$-value of less than .05 will be considered statistically significant.

## Results

### Respondents' characteristics

1,297 respondents participated in this study. The overview of respondents' socio-demographic background is in Table 2. Majority of the respondents were female (59.2%), below the age of 40 (62.9%), have some tertiary education background (89.3%), Muslim (43.7%), lived in a city (61.1%) and Klang Valley (74%) and perceived themselves to be earning more than the national monthly median income of RM2,442 or US$580 (48.7%).

Upon further categorization of the respondents, the majority reported that they intended to be vaccinated by registering in the national vaccination program PICK, or had received at least one dose of vaccination (92.2%). Table 2 contains the overview of respondents' socio-demographic background and their vaccination intention.

98.7% of respondents reported having mySejahtera on their phone, indicating that vaccination registration is frictionless for them. Vaccination registration to the public opened in late February 2021 and the majority of our respondents reported to have registered immediately (56.4%). Among respondents who had registered, 72.5% of registrations happened in February and March 2021. These results pointed to a strong pro-vaccination attitude prevalent among respondents, and the frictionless registration process assisting public registration. Upon further examination, 63.0% of individuals who had registered for vaccination had received at least 1 dose indicating that individuals turned their intentions into desirable actions.

### Vaccination expectations

In Block 2, respondents reported their beliefs pertaining to the share of individuals who intended to get vaccinated. We used this as a measure of empirical expectations on pro-vaccination norms among our sample. Fig 2 represents the results of this measure by registration status.

From Fig 2, we found that respondents who had reported vaccination registration on average expected more individuals in their social groups to register for vaccination. The expectations on vaccination were highest among registered individuals towards individuals in the same household and among family members. Among registered respondents, expectations on registration declined as size of the social groups grew and included more strangers. These

**Table 2. Demographics and COVID-19 vaccination registration status.**

| | | Intention to be vaccinated | |
|---|---|---|---|
| | | Registered in PICK | Not Registered |
| | Overall | n = 1196 | n = 101 |
| | N (%) | n (%) | n (%) |
| Gender | | | |
| Female | 756 (59.2) | 694 (91.8) | 62 (8.2) |
| Male | 521 (40.8) | 483 (92.7) | 38 (7.3) |
| Age group (years) | | | |
| 18–30 | 498 (38.4) | 456 (91.6) | 42 (8.4) |
| 31–40 | 317 (24.5) | 289 (91.2) | 28 (8.8) |
| 41–50 | 192 (14.8) | 176 (91.7) | 16 (8.3) |
| 51–60 | 177 (13.7) | 164 (92.7) | 13 (7.3) |
| >60 | 113 (7.3) | 111 (98.2) | 2 (1.8) |
| Highest education level | | | |
| Secondary and below | 139 (10.7) | 126 (90.6) | 13 (9.4) |
| Tertiary | 1158 (89.3) | 1070 (92.4) | 88 (7.6) |
| Religion | | | |
| Islam | 567 (43.7) | 524 (92.4) | 43 (7.6) |
| Buddhism | 293 (22.6) | 268 (91.5) | 25 (8.5) |
| Christianity | 190 (14.7) | 177 (93.2) | 13 (6.8) |
| Others | 129 (9.9) | 118 (91.5) | 11 (8.5) |
| None | 118 (9.1) | 109 (92.4) | 9 (7.6) |
| Residential area | | | |
| Urban | 1192 (91.9) | 1109 (93.0) | 83 (7.0) |
| Rural | 105 (8.1) | 87 (82.9) | 18 (17.1) |
| Current residence | | | |
| East Malaysia | 56 (4.3) | 51 (91.1) | 5 8.9) |
| Klang Valley | 960 (74.0) | 897 (93.4) | 63 (6.6) |
| Northern Peninsular | 118 (9.1) | 103 (87.3) | 15 (12.7) |
| Southern Peninsular | 95 (7.3) | 86 (90.5) | 9 (9.5) |
| East Peninsular | 68 (5.2) | 59 (86.8) | 9 (13.2) |
| Perceived Income from Median Income | | | |
| Extremely below median | 113 (8.7) | 103 (91.2) | 10 (8.8) |
| Below median | 142 (11.0) | 116 (81.7) | 26 (18.3) |
| Close to median | 329 (26.9) | 316 (96.0) | 33 (10.0) |
| Above median | 604 (46.7) | 576 (95.4) | 28 (4.6) |
| Extremely above median | 89 (6.7) | 85 (95.5) | 4 (4.5) |

Current residence combined states into several regions. East Malaysia consisted of Sarawak, WP Labuan, and Sabah. Klang Valley consisted of Selangor, WP Kuala Lumpur, and WP Putrajaya. Northern Peninsular Malaysia consisted of Perak, Kedah, Pulau Pinang, and Perlis. Southern Peninsular Malaysia consisted of Negeri Sembilan, Melaka, and Johor while Eastern Peninsular Malaysia consisted of Pahang, Terengganu, and Kelantan.

results hint that respondents' pro-vaccination behaviour can be linked to the expectations of other members of the relevant social group also having pro-vaccination behaviour. Summary statistics and statistical differences in expectations among unregistered and registered respondents can be found in Table 3. Results from Table 3 show there are statistical differences in registered and unregistered respondents' vaccination expectations. For different types of social

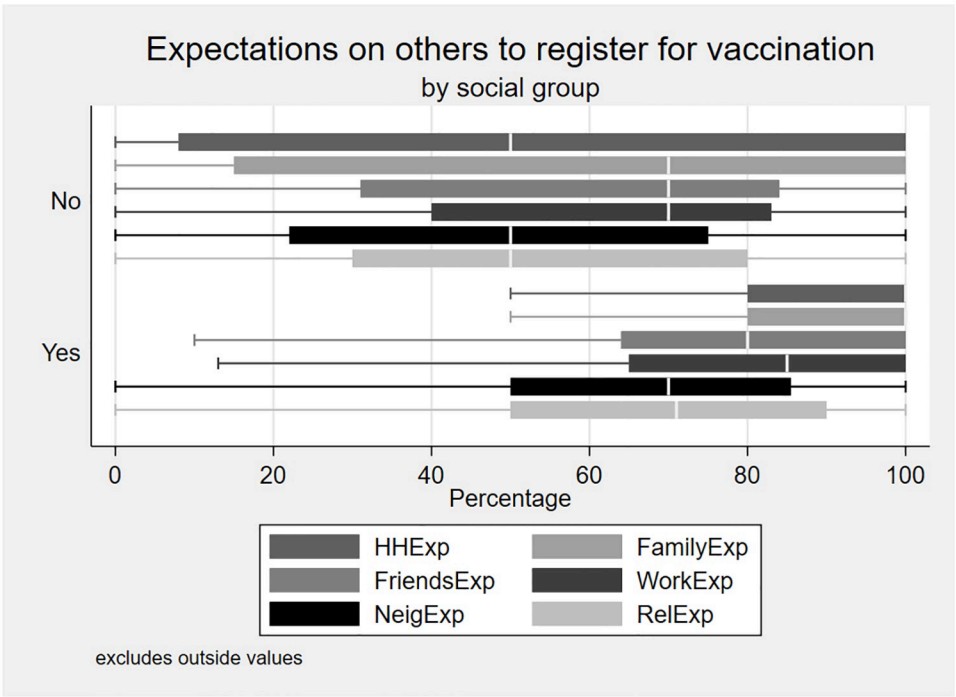

**Fig 2. Expectations on others to register for vaccination by social groups.** Respondents are categorized as those who had yet to register for vaccination (*No*), and those who had registered for vaccination (*Yes*). We excluded outside values in the figure above. *HHExp*: Expectations within the same household, *FamilyExp*: Expectations among family members, *FriendsExp*: Expectations among friends, *WorkExp*: Expectations among neighbors, *RelExp*: Expectations towards individuals in the same religion.

groups, respondents who had registered for vaccination are on average more likely to expect other individuals in their reference group to be registered and willing to be vaccinated.

Respondents were asked to estimate vaccination registration within their state of residence's population but results from Table 3 show that on average there is no statistical difference in

**Table 3. Summary statistics and statistical differences in expectations that others will register for vaccination.**

| Social groups | Registered PICK | | Unregistered PICK | | Differences |
|---|---|---|---|---|---|
| | Mean | SD | Mean | SD | (q-values) |
| Household members | 84.5 | 28.7 | 52.2 | 39.8 | 9.478*** |
| Family members | 83.1 | 27.5 | 58.1 | 38.6 | 7.982*** |
| Friends | 76.0 | 24.6 | 58.5 | 31.8 | 6.060*** |
| Co-workers | 77.0 | 25.7 | 60.9 | 31.6 | 4.539*** |
| Neighbours | 65.3 | 26.4 | 51.1 | 31.3 | 4.158*** |
| People with the same religion | 68.0 | 26.0 | 50.7 | 30.0 | 3.776*** |
| People in the same state | 57.8 | 25.3 | 53.2 | 24.8 | 1.767 |

* $q < 0.10$,

** $q < 0.05$,

*** $q < 0.01$

*Registered PICK*: Respondents who had registered for vaccination, *Unregistered PICK*: Respondents who had not registered for vaccination. Mean and standard deviations (SD) are calculated from the percentage of individuals within certain social groups who will register for vaccination.

**Table 4. Pair-wise correlations of vaccination registration expectations across social groups.**

| | Household | Family | Friends | Co-workers | Neighbors | Religions |
|---|---|---|---|---|---|---|
| **Registered with PICK** | | | | | | |
| Household | | | | | | |
| Family | 0.759*** | | | | | |
| Friends | 0.529*** | 0.633*** | | | | |
| Co-workers | 0.470*** | 0.536*** | 0.700*** | | | |
| Neighbours | 0.460*** | 0.548*** | 0.686*** | 0.634*** | | |
| Religions | 0.415*** | 0.443*** | 0.610*** | 0.617*** | 0.696*** | |
| State | 0.1703*** | 0.146*** | 0.226*** | 0.173*** | 0.179*** | 0.141*** |
| **Unregistered with PICK** | | | | | | |
| Household | | | | | | |
| Family | 0.843*** | | | | | |
| Friends | 0.621*** | 0.672*** | | | | |
| Co-workers | 0.537*** | 0.541*** | 0.746*** | | | |
| Neighbours | 0.468*** | 0.537*** | 0.842*** | 0.540*** | | |
| Religions | 0.318* | 0.344* | 0.703*** | 0.582*** | 0.783*** | |
| State | 0.322*** | 0.306*** | 0.489*** | 0.326** | 0.454*** | 0.576*** |

* $p < 0.10$,

** $p < 0.05$,

*** $p < 0.01$

registration expectation between unregistered and registered respondents. The expectations of vaccination among others living in the same state as the respondents are the lowest among the other social groups. This implies that some social groups matter more than others in normalizing pro-vaccination behaviour. For registered respondents, the low expectation of vaccination registration could reflect their the strength of norms operating within a smaller social group and/or their personal normative beliefs that vaccination is the right behaviour irrespective of what others are doing. On the other hand, those who choose not to register do it because they expect that others within their social groups have not registered, expecting that their personal normative beliefs are being upheld by a huge share of their social groups.

Table 4 contains the pair-wise correlations between different social groups by their vaccination registration status. Results from Table 4 shows that expectations of smaller social circles i.e. households, family members and friends are strongly correlated with each other for both unregistered and registered individuals.

To address the relationship of personal normative beliefs on vaccination, we asked the respondents whether vaccinations produce personal benefits. Personal benefits here are a direct effect on self and others in the same social and economic circles. The majority of our respondents believed vaccinations were personally beneficial (97%) including majority of unregistered respondents (82%). While majority of unregistered respondents believe that vaccinations are beneficial, statistically this is lower that the belief registered by registered individuals (t-test, t = 4.3139, p-value < 0.01).

Internalizing pro-vaccination norms manifest not only in high vaccination levels within a population, but also the willingness of individuals to uphold those norms by sanctioning or excluding norm violators. The sub-sample who had reported they have registered for vaccination were asked what their response was when they had to interact with individuals whom they

know refused to be vaccinated. The available options for this were, i) status quo inline with government directive (maintain social distancing and mask), ii) *engage in extra effort to avoid these individuals, like eating alone OR avoid sharing a bathroom with them OR avoid sending my kids to schools/kindergarten if they have kids in the same place*, and iii) normalizing pre-pandemic behaviour (*I feel confident not to wear a mask around them*). The second option, i.e. engaging in extra effort to avoid unvaccinated individuals, is the proxy of sanctioning behaviour for norms violators. Avoidance by affected second party requires personal cost- be it psychological or economic- as they must engage in extra mental effort to be vigilant, or filter for potentially beneficial interactions.

Among our respondents, 76.5% indicated that they would continue with the status quo, 19.4% indicated they would engage in extra effort to avoid un-vaccinated individuals, while the remaining 4.1% indicated that they were ready to return to pre-pandemic social interaction without masks. Additional analysis found that individuals' vaccination expectations had no relationship with their reactions when dealing with un-vaccinated individuals. Since the status quo option is consistent with the contemporaneous masking mandate by the government, there is the possibility that complying with social distancing regulations crowds out individuals' actions to engage in any type of sanctioning behaviour towards un-vaccinated individuals, i.e. continuing masking with vaccination provided perceived protection against potential negative side effects from interacting with un-vaccinated individuals.

## Trust and expectations on government

Individuals' pro-vaccination norms are facilitated with strong government commitments to deliver those shots in citizens' arms in a timely manner. Trust levels are found to facilitate policy implementations as a high level of trust in the society and towards policy actors are positively correlated to compliance towards social distancing measures [45] and vaccination rates [46] in society. We replicate the generalized trust question from World Value Survey in our survey. Overall, 73% of our respondents reported that they needed to be very careful in dealing with others. In comparison, 80.4% of samples in the latest wave of World Values Survey reported that they don't trust others. While the majority of our samples have low generalized trust, unregistered respondents reported a statistically higher proportion of the need to be very careful in dealing with people (t-test, t = 2.1715, p-value <0.05).

Table 5 summarized respondents' trust levels towards government actors based on their vaccination registration status. Their answers were categorised into a 4-level Likert scale, namely; *i) "don't trust at all", ii)"do not trust very much", iii) "trust somewhat", and iv) "trust completely"*. These actors were selected as they had the ability to influence vaccination take-up in the population through effective delivery systems (Ministry of Health (MOH) and medical

**Table 5. Levels of trust towards policy actors by vaccination registration status.**

|  |  | Do not trust at all | Do not trust very much | Trust somewhat | Trust completely |
|---|---|---|---|---|---|
| Elected representative | Registered | 19.5% | 40.2% | 39.2% | 5.1% |
|  | Unregistered | 34.5% | 27.6% | 32.2% | 5.8% |
| Ministry of Health | Registered | 3.0% | 16.0% | 51.4% | 30.0% |
|  | Unregistered | 15.8% | 22.8% | 38.6% | 22.8% |
| JAKIM | Registered | 27.2% | 22.6% | 30.9% | 19.3% |
|  | Unregistered | 34.9% | 24.1% | 24.1% | 16.8% |
| Medical doctors | Registered | 0.6% | 4.6% | 48.3% | 46.5% |
|  | Unregistered | 4% | 18.2% | 45.5% | 32.2% |

doctors) and persuasion (JAKIM, elected representatives). Trust towards JAKIM is included here as it is a federal agency that is responsible to certify products as *halal* or permissible for Muslim consumption, and it is therefore assumed to have a high persuasive influence on Muslims to take up COVID-19 vaccination. On the 23rd of December 2020, JAKIM announced the COVID-19 vaccine permissible for Muslim consumption.

Table 5 shows that registered respondents had greater trust towards policy actors in comparison to unregistered respondents. This is applicable to trust directed toward elected representatives and MOH. Irrespective of vaccination registration status, low trust levels were reported towards JAKIM, while trust towards medical doctors was highest. The most polarizing trust gap can be found when it was directed towards the MOH, i.e. the leading Ministry that manages pandemic responses, including vaccine certification and mobilizing of human resources.

At the start of vaccination roll-out programs, Malaysia set a target that 80% of its adult population needed to be vaccinated to achieve herd immunity. As there is no scientific consensus on the threshold of herd immunity, we asked our respondents to estimate the percentage of population in their state that needed vaccination in order to reach herd immunity, relaying their expectations towards other Malaysians and government capability to deliver the vaccine. According to them, the average percentage of population that needed to be vaccinated was 81.43%. However, upon closer examination, respondents yet to be registered under PICK on average reported vaccinated percentage should be 68.93%, while registered individuals expected that 82.49% of their state population should get vaccinated in order to achieve herd immunity. The differences between herd immunity expectations of un-registered and registered respondents are statistically significant (unequal t-test, t = 4.230, p-value <0.001). There are several probable explanations in the differences between registered and un-registered respondents. Low expectations on herd immunity could be interpreted by what respondents believe. If they believe herd immunity can be reached at a low threshold, then they believe that there are free riders, i.e. individuals that choose not to get vaccinated and rely on others getting the vaccine. Low expectations also could stem from refusing to believe the existence of herd immunity. In this case, the government's herd immunity goal has no implication on individuals' voluntary actions.

Governments have the ability to implement policies to increase vaccination inline with policy targets. Normalization of vaccination within a population implies that government's actions towards un-vaccinated segments of the population have support from the general population, even if the implementation of these policies are costly and time-consuming. Rather than querying individuals' support for these policies, we elicited respondents' expectations of the percentage of population that supported two types of policies which could increase vaccination take-up; i) mandates and punishment for refusers, and ii) increased accessibility and incentives for vaccination. Overall, respondents expected that government's policies in the realm of accessibility and incentives have greater societal support in comparison to policies that relied on mandates and punishment. The average support for incentives were at 77% and for punishment were 60% (t-test, t = 22.2742, p-value < 0.001). Among respondents who had not registered their interest to be vaccinated, average expectations on any type of government policies' societal supports were significantly lower than those who had registered. Fig 3 illustrates respondents' expectations on government policies by their registration status.

Fig 3 ties individuals' expectations with government's actions. Actions by the government to increase COVID-19 vaccination take-up reinforce personal normative beliefs that vaccination is the correct behavior in the society and this is true for individuals that have taken the vaccine or will take the vaccine. At the same time, holding high expectations that societal support for pro-vaccination policy strengthen the notion that there is high pro-vaccination norms

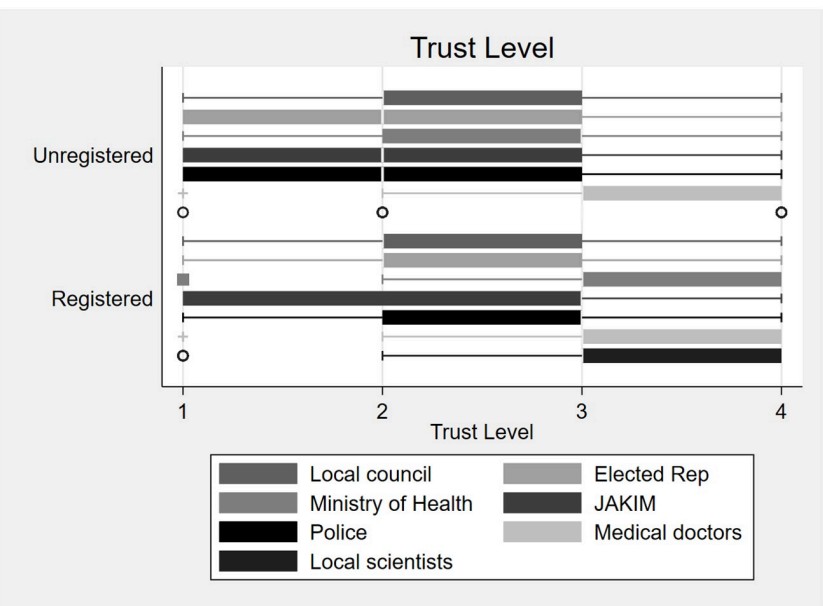

**Fig 3. Expectations on population support towards policies designed to increase COVID-19 vaccination take-up by respondents' registration status.** At this stage of the survey, we did not specify policy tools that would serve as punishment or incentives. We also did not distinguish between monetary or intrinsic incentives, so respondents were free to assume any positive incentive that they thought effective for vaccination.

within a community along with willingness to see public resources being used to achieve government's target.

## Vignette analysis

We return to the hypotheses listed above and examine respondents' beliefs on the unregistered protagonist's: i) likelihood to register to be vaccinated, ii) likelihood to receive vaccination and iii) should the protagonist get vaccinated. Respondents are randomly sorted into a situation in which majority residents in Malaysia have been registered (or refuse to register) for vaccination, and vaccination is either incentivised through a nationwide lottery or vaccine refusal is dis-incentivised through frequent PCR Covid tests. HP conditions in the vignette imply a high level of normative expectations within the population that vaccination is the approved action, while the LP reflects a low level of normative expectations on vaccination take-up. The prevalence of the high/low vaccination take-up within a community could happen in the context of government incentivizing (LOT) the population, or disincentivizing vaccine refusal (PUN).

Table 6 shows the breakdown of respondents randomly assigned in each treatment group.

Each respondent reported three types of expectations and each in turn reflected different stages of vaccination decision-making. First, respondents reported their expectations on the protagonist's intention to be vaccinated. Second, a response to the likelihood that the protagonist would follow through with their intention and receive the vaccination. We distinguised intention with actual behavior as during the time of this survey roll out, there was a concern that some segments of society refused to be vaccinated but still registered for vaccination as a way to avoid potential social exclusion. The third type of expectation reported on respondents' projections of their personal normative beliefs towards the protagonist, i.e. should the protagonist take the vaccination.

**Table 6. Breakdown of respondents by vignette treatments.**

|  | LP | HP | Total |
|---|---|---|---|
| LOT | 320 | 337 | 657 |
|  | (25.7) | (27.1) | 52.8 |
| PUN | 286 | 301 | 587 |
|  | (23.0) | (24.2) | 47.2 |
| Total | 606 | 638 | 1244 |
|  | (48.7) | (51.3) | (100.0) |

Percentage of total respondents in parentheses

Table 7 shows differences in respondents' expectations based on the stages of protagonist's vaccination decisions. Specification (1) conditions the norm comparison on a lottery, while (2) conditions it on a punitive government policy. For specifications (1) and (2), mean responses to the low registration rate vignette are subtracted from the mean response to the high registration rate vignette, capturing the effect of pro-registration expectations on these hypothetical decisions. (3) conditions the incentive valence on high registration expectations, while (4) conditions it on low registration expectations. For specifications (3) and (4), mean responses to a lottery are subtracted from mean responses to the self-testing at own expense punishment, capturing the relative effect of a negative incentive as compared to a positive one on these hypothetical decisions.

**H1** hypothesised that respondents assigned to HP treatments are more likely to expect their protagonist to be willing to register for vaccination and/or translate this into actual vaccination than those assigned to the LP treatments. However, we see that the social expectation manipulation has relatively little effect on responses. None are significant at the $\alpha = 0.05$, so we reject H1 as shown in (1) and (2) of Table 7. This is a counter-intuitive result, so we may wonder whether high expectations truly don't matter, or if the effect size is simply too small to be detected with our vignette experiment. We present evidence suggesting the latter in S2 Appendix.

On the other hand, respondents are more likely to believe that the protagonist will take actions when we analyse the effectiveness of LOT and PUN by varying vaccination

**Table 7. Effect of incentives and expectations on hypothetical vaccination behavior–T-tests.**

|  | LOT:HP vs LP | PUN:HP vs LP | HP:LOT vs PUN | LP:LOT vs PUN |
|---|---|---|---|---|
|  | (1) | (2) | (3) | (4) |
| Will register | -0.0966* | 0.0242 | 0.00121 | 0.122** |
|  | (0.0560) | (0.0578) | (0.0557) | (0.0581) |
| Will get vaccinated | -0.0671 | 0.0506 | 0.0870 | 0.205*** |
|  | (0.0590) | (0.0573) | (0.0573) | (0.0596) |
| Should be vaccinated | -0.0138 | -0.0823* | 0.267*** | 0.198*** |
|  | (0.0530) | (0.0427) | (0.0472) | (0.0505) |
| Observations | 657 | 587 | 638 | 606 |

Standard errors in parentheses

* $p < 0.10$,

** $p < 0.05$,

*** $p < 0.01$

expectations in the community (HP and LP). Columns (3) and (4) address the **H2** prescribed in the earlier section. Column (3) inferred that in a community with high expectations (HP) that others have registered and are willing to be vaccinated, there is no significant difference in the relative efficacy of LOT and PUN. But, respondents are likely to believe LOT is more effective in persuading the protagonist to register compared to PUN. However, the relative efficiency of LOT is more apparent among respondents assigned in the community with low expectation on vaccination (LP), as shown in column (4) of Table 7. Given the likelihood of respondents assigning low importance of expectation variations shown in Columns (1) and (2), we reject **H2** as well. Regardless, LOT is has a relative advantage over PUN under LP treatments.

Table 8 aims to explain the effect of incentives, norms and beliefs on the protagonist's expected behavior. Under the PUN, low trust is strongly associated with LP and vaccination, but has no significant relationship with norms. Under LOT manipulation, by contrast, low trust has no significant relationship with behavior or norms, and therefore the relative efficacy of LOT in driving behavior is significantly higher for low-trust individuals relative to high-trust individuals. The generally lower efficacy of LOT on behavior is almost entirely driven by high-trust respondents. Thus, while we have strong evidence of the behavioral components that support pro-vaccination, government intervention has a positive relationship with protagonist's expected behavior. However, we found no evidence that expectations of social pro-vaccination behaviour will project into the likelihood of protagonists adjusting their behavior and getting the vaccine.

**Table 8. Effect of incentives, norms, and beliefs on hypothetical vaccination behavior.**

|  | Will register | Will get vaccinated | Should get vaccinated |
|---|---|---|---|
|  | (1) | (2) | (3) |
| LOT | -0.181*** | -0.250*** | -0.121*** |
|  | (0.0489) | (0.0507) | (0.0450) |
| low trust | -0.127*** | -0.170*** | -0.00199 |
|  | (0.0412) | (0.0427) | (0.0379) |
| LOT × low trust | 0.167*** | 0.195*** | -0.0848 |
|  | (0.0576) | (0.0597) | (0.0530) |
| HP | 0.0268 | 0.00189 | 0.0336 |
|  | (0.0257) | (0.0267) | (0.0237) |
| Registered | 0.178** | 0.111 | 0.296*** |
|  | (0.0691) | (0.0722) | (0.0636) |
| HH expectation | 0.000200 | -0.0000846 | 0.000920** |
|  | (0.000473) | (0.000490) | (0.000435) |
| Live alone | -0.0251 | -0.0320 | -0.00625 |
|  | (0.0546) | (0.0566) | (0.0502) |
| Support intervention | 0.00132*** | 0.00173*** | 0.00131*** |
|  | (0.000342) | (0.000355) | (0.000315) |
| Observations | 1224 | 1223 | 1224 |

Standard errors in parentheses

*$p < 0.10$,

**$p < 0.05$,

***$p < 0.01$

* Support Intervention is the average of perceived support for rewards and support punishment in the respondent's state

We see that Household registration expectations (HH Expectation) have no significant effect on protagonist behavior, but significantly influence respondent norms. "Support Intervention" is derived by averaging the two social expectations: respondent's expectation of percentage popular support within their state, 0–100, and government registration interventions using rewards and punishments, respectively. We see that these beliefs have a large and very significant effect on both protagonist behavior and norms. Table 8 shows there is very little of the variation in these beliefs is explained by the respondent's state of residence–variations in these beliefs are almost entirely due to individual idiosyncrasies in social expectations, and state-level omitted variables cannot explain their relationship with vignette outcomes.

## Testing whether lottery able to incentivize registration

We examined the effectiveness of lottery and offered a segment of unregistered respondents the chance to win RM500 (USD 126) if they registered for vaccination before a deadline. 53 unregistered respondents were assigned to the lottery experiment: 33 to treatment arm and 20 to control. Two in the treatment arm and one in the control provided invalid phone numbers and were dropped from the analysis. For the remainder, response to the treatment was measured by registration status one week after the close of the survey, queried from the public PICK registration database using the phone number provided. 26 of the 31 respondents offered the lottery were registered within one week of the closing of the survey, while only 12 of the 19 respondents in the control registered in that time-frame. This suggests a large effect size: just 17% of the treated remained unregistered compared with 37% of the untreated. Due to the small sample size, we used Boschloo's Exact Test. We performed a one-tailed test of whether the proportion of treated who registered was higher than the proportion of untreated. The p-value is 0.056, so we cannot reject the null hypothesis at $\alpha = 0.05$. We did not do a follow-up to determine if every unregistered respondent eventually received vaccination or not, as their personal details were destroyed after this lottery was concluded, and the government website that allowed individuals to verify registration and vaccination status was deactivated by January 2022.

## Discussion and limitations

Our survey shows that respondents' own registration behavior is strongly correlated with that of their household, and progressively less correlated with more distant social connections like friends, neighbors, and members of their religion. This may reflect both assortation–like people cluster together–and the power of social norms in driving vaccination behavior.

The vignette experiment suggests empirical expectations have little effect on vaccination behavior, at least in the hypothetical. Rewarding protagonists are believed to be much more effective in incentivizing pro-vaccination behavior, particularly among respondents who exhibit lower generalized trust. However, trust has no relationship with respondents' opinion of whether the protagonist should vaccinate.

These results are surprising, and we hypothesize that since respondents may view the protagonist not just as a proxy for themselves, but also as a representative Malaysian, low trust may influence protagonist behavior via an additional channel: distrust in the protagonist. Then we could interpret these results as saying that both high– and low– trust individuals see government punishment as obligating greater compliance, but only high–trust individuals actually believe that protagonists will follow through with that obligation. We provide evidence for this hypothesis in S2 Appendix under S5 and S6 Tables). Further suggesting that the protagonist may be seen as a representative Malaysian, respondents with higher household registration rates are more likely to think the protagonist should register, but not that they will

register. Finally, while our experimentally–assigned empirical expectations have little effect, respondents' real–world normative expectations of support for government intervention in vaccination are strongly associated with the belief that protagonists should and will vaccinate.

While a lottery and threat of punishment in the vignette experiment have no statistical significant implication on our respondents' beliefs, our small lottery experiment suggests that even lotteries greatly decreased the number of individuals refusing the register for the vaccine, though the small sample size makes this effect only marginally significant.

These findings are particularly important for any future vaccination policy. Apart from ensuring equitable distribution of COVID-19 vaccination, a challenge to policymakers is to ensure there are enough arms to take up the vaccination. As of August 2022, the world's population vaccination rate stands at 61.7%. Full vaccination coverage in countries in Africa and Caribbean regions are still low, with less than 10% of the populations of Burkina Faso and Haiti having received vaccination [47–49]. With growing evidence of waning effectiveness, countries are beginning to offer additional COVID-19 vaccines to their population as booster [50, 51]. Even countries like Malaysia are now struggling to convince its population to take-up COVID-19 boosters and obtain permission from parents and guardians to enable children's vaccinations, since there is no policy to incentivize booster shot(s) or disincentivize refusal of take-up. Social norm, either driven by empirical or normative expectations, might play an important role to normalize pro-vaccination norms once policy interventions are removed, or perceived clear-cut benefits from vaccination are no longer apparent.

## Limitations

We acknowledge that this work has some limitations. First, the vignette incentives that we explored had no links to the Malaysian government's eventual policy actions. For example, in October 2021, there was a vaccination mandate for civil servants, and several social activities like indoor dining and the use of sports facilities. We can establish that the loosening of restrictions and exclusion of unvaccinated individuals was the result of vaccination policy decisions of Malaysia at the end of 2020. Administratively, by November 2021, the government had decided to abandon its herd immunity target, and pushed to vaccinate everyone [52]. This mandate and exclusion from daily activities might have influenced Malaysians get vaccinated. As of March 2022, over 84% of eligible residents of Malaysia had received two doses of the COVID-19 vaccine.

Secondly, this work only captured vaccination decisions before the biggest hospitalization and death wave in Malaysia, which was due to the Delta wave. Peaks in hospitalizations and deaths happened a month after this survey was completed. This survey also did not capture government responses like increasing vaccination sites, and more aggressive media campaigns to persuade the public to get vaccinated. The negative outcomes stemming from unvaccinated individuals who fell sick or died, along with the greater availability of vaccinations, may have convinced hesitant individuals to get vaccinated. Since we did not plan for any follow-up surveys involving this set of respondents, we were not able to determine whether unregistered individuals ended up getting vaccinated, and their reasons for changing their minds.

## Supporting information

**S1 Appendix. Survey questions.** Survey questions used on Qualtrics.
(PDF)

**S2 Appendix. Supplementary analyses.**
(PDF)

## Acknowledgments

The authors would like to thank Tan Zhai Gen, Kar Yern Chin, Shenyi Chua, and Fatin Nadhirah Jamalolail for their assistance in the data collection process for this project.

## Author Contributions

**Conceptualization:** N. Izzatina Abdul Aziz, Sam Flanders, Melati Nungsari.

**Data curation:** N. Izzatina Abdul Aziz, Sam Flanders, Melati Nungsari.

**Formal analysis:** N. Izzatina Abdul Aziz, Sam Flanders, Melati Nungsari.

**Funding acquisition:** N. Izzatina Abdul Aziz, Sam Flanders, Melati Nungsari.

**Investigation:** N. Izzatina Abdul Aziz, Sam Flanders, Melati Nungsari.

**Methodology:** N. Izzatina Abdul Aziz, Sam Flanders, Melati Nungsari.

**Project administration:** N. Izzatina Abdul Aziz, Sam Flanders, Melati Nungsari.

**Resources:** N. Izzatina Abdul Aziz, Sam Flanders, Melati Nungsari.

**Software:** N. Izzatina Abdul Aziz, Sam Flanders, Melati Nungsari.

**Supervision:** N. Izzatina Abdul Aziz, Sam Flanders, Melati Nungsari.

**Validation:** N. Izzatina Abdul Aziz, Sam Flanders, Melati Nungsari.

**Visualization:** N. Izzatina Abdul Aziz, Sam Flanders, Melati Nungsari.

**Writing – original draft:** N. Izzatina Abdul Aziz, Sam Flanders, Melati Nungsari.

**Writing – review & editing:** N. Izzatina Abdul Aziz, Sam Flanders, Melati Nungsari.

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
