## [Decision Letter · Decision Letter 0]

3 Aug 2022

PONE-D-22-18292Social expectations and government incentives in Malaysia's COVID-19 vaccination uptakePLOS ONE

Dear Dr. Nor Izzatina Abdul Aziz

Thank you for submitting your manuscript to PLOS ONE. After careful consideration, we feel that it has merit but does not fully meet PLOS ONE’s publication criteria as it currently stands. Therefore, we invite you to submit a revised version of the manuscript that addresses the points raised during the review process.

ACADEMIC EDITOR: Minor revision addressing reviewers comments to be done and submitted.

We look forward to receiving your revised manuscript.

Kind regards,

Srikanth Umakanthan

Academic Editor

PLOS ONE

Journal Requirements:

Additional Editor Comments:

Minor revision addressing the reviewers comments to be done by the authors.

Reviewers' comments:

Reviewer's Responses to Questions

**Comments to the Author**

1. Is the manuscript technically sound, and do the data support the conclusions?

Reviewer #1: Yes

Reviewer #2: Yes

2. Has the statistical analysis been performed appropriately and rigorously? 

Reviewer #1: Yes

Reviewer #2: N/A

3. Have the authors made all data underlying the findings in their manuscript fully available?

Reviewer #1: Yes

Reviewer #2: Yes

4. Is the manuscript presented in an intelligible fashion and written in standard English?

Reviewer #1: Yes

Reviewer #2: Yes

5. Review Comments to the Author

Reviewer #1: 1. Abstract- Well-structured and summarizes the overall purpose of the study and the research problem(s) investigated. The basic design of the study; major findings and trends found because of the study is also showcased.

2. Materials and methods- The authors have included a proper survey questionnaire along with the assessment scale to analyze their objectives. The settings is very elaborated, requires significant tapering of the settings and survey design.

3. Statistics: The involvement of descriptive analysis in the study has being well established.

4. Discussion: The discussion requires more additional statements with regards to the existing literature search. Include the following references and citations reflecting the COVID-19 updates to strengthen the manuscript:

- Origin and transmission (use reference and cite: doi:10.1136/postgradmedj-2020-138234)

- To mention in brief about vaccines (use reference and cite: “doi: 10.1136/postgradmedj-2021-141365.

AND doi:10.1136/postgradmedj-2021-140654”)

- Definition of vaccine resistance and hesitance (use reference and cite: doi:10.3390/vaccines9101064).

- Compare the global vaccine status and relate it with the current vaccine status (refer and cite: doi: 10.3389/fpubh.2022.844333)

5. Conclusion: The authors have shown the importance of Social Expectations and Government Incentives in Malaysia’s COVID-19 Vaccine Uptake.

I advocate this article for revision pending inclusion of the points as recommended by me.

Reviewer #2: The manuscript depicts a technically sound piece of scientific research with the data supporting the conclusion, indicating it has been rigorously researched and presented in an intelligible fashion.

6. PLOS authors have the option to publish the peer review history of their article (what does this mean?). If published, this will include your full peer review and any attached files.

Reviewer #1: No

Reviewer #2: No

---

## [Author Response · Author response to Decision Letter 0]

7 Sep 2022

As suggested by Reviewer 1, we have included relevant statements that touch issues mentioned above to improve our manuscript. The additional statements relevant to your suggestions are underlined with a short note on the margin. The suggested and related references have included in the References section. In the marked-up revised manuscript this can be identified in blue font.

---

## [Editor Report · Decision Letter 1]

9 Sep 2022

Social expectations and government incentives in Malaysia's COVID-19 vaccine uptake

PONE-D-22-18292R1

Dear Dr. Nor Izzatina Abdul Aziz,

We’re pleased to inform you that your manuscript has been judged scientifically suitable for publication and will be formally accepted for publication once it meets all outstanding technical requirements.

Kind regards,

Srikanth Umakanthan

Academic Editor

PLOS ONE

Additional Editor Comments (optional):

The authors have addressed the comments of the reviewers. The revised manuscript is now accepted in the present format.
---

## [Editor Report · Acceptance letter]

15 Sep 2022

PONE-D-22-18292R1 

Social expectations and government incentives in Malaysia's COVID-19 vaccine uptake 

Dear Dr. Abdul Aziz:

I'm pleased to inform you that your manuscript has been deemed suitable for publication in PLOS ONE. Congratulations! Your manuscript is now with our production department. 

Kind regards, 

on behalf of

Dr. Srikanth Umakanthan 

Academic Editor

PLOS ONE